# Effect and feasibility of district level scale up of maternal, newborn and child health interventions in Pakistan: a quasi-experimental study

Zahid Ali Memon ![ORCID] ,[1] Shah Muhammad,[1] Sajid Soofi ![ORCID] ,[1] Nimra Khan,[1] Nadia Akseer,[2] Atif Habib,[1] Zulfiqar Bhutta[1,2]

[1]Centre of Excellence in Women and Child Health, Aga Khan University, Karachi, Pakistan
[2]Centre for Global Child Health, The Hospital for Sick Children, Toronto, Ontario, Canada

**Correspondence to**
Professor Zulfiqar Bhutta;
zulfiqar.bhutta@aku.edu

## ABSTRACT

**Introduction** Pakistan has a high burden of maternal, newborn and child morbidity and mortality. Several factors including weak scale-up of evidence-based interventions within the existing health system; lack of community awareness regarding health conditions; and poverty contribute to poor outcomes. Deaths and morbidity are largely preventable if a combination of community and facility-based interventions are rolled out at scale.

**Methods and analysis** Umeed-e-Nau (UeN) (New Hope) project aims is to improve maternal, newborn and child health (MNCH) in eight high-burden districts of Pakistan by scaling up of evidence-based interventions. The project will assess interventions focused on, first, improving the quality of MNCH care at primary level and secondary level. Second, interventions targeting demand generation such as community mobilisation, creating awareness of healthy practices and expanding coverage of outreach services will be evaluated. Third, we will also evaluate interventions targeting the improvement in quality of routine health information and promotion of use of the data for decision-making. Hypothesis of the project is that roll out of evidence-based interventions at scale will lead to at least 20% reduction in perinatal mortality and 30% decrease in diarrhoea and pneumonia case fatality in the target districts whereas two intervention groups will serve as internal controls. Monitoring and evaluation of the programme will be undertaken through conducting periodical population level surveys and quality of care assessments. Descriptive and multivariate analytical methods will be used for assessing the association between different factors, and difference in difference estimates will be used to assess the impact of the intervention on outcomes.

**Ethics and dissemination** The ethics approval was obtained from the Aga Khan University Ethics Review Committee. The findings of the project will be shared with relevant stakeholders and disseminated through open access peer-reviewed journal articles.

**Trial registration number** NCT04184544; Pre-results.

## Strengths and limitations of this study

► Strength of the Umeed-e-Nau project includes its quasi-experimental study design with district representative sample size.
► The household surveys will be powered to measure district level perinatal and child mortality estimates.
► Interventions to promote use of data for decision-making will be assessed in Pakistan for the first time.
► Potential limitations include challenges in accessing routine service delivery data for evaluation purposes.
► Challenges of working with or assessing role of lady health workers who are already overburdened with competing priorities.

maternal mortality ratio in 2017 was estimated at 211 maternal deaths per 100 000 live births while 2019 estimates by United Nations inter-agency group for child mortality estimation (UN-IGME) estimate neonatal mortality rate at 18 deaths per 1000 live births and under-5 mortality rate is 39 per 1000 live births per year globally.[2 3] Diarrhoea and pneumonia are two of the most common causes of child mortality; together, they account for more than 30% of all under-5 deaths in Pakistan.[4]

Several factors have contributed to the slow progress in maternal, newborn and child health (MNCH) in Pakistan, including social determinants such as poverty, high rates of illiteracy, lack of women's and girls' empowerment, voice, and high fertility rates. These are compounded by poorly functional health systems with low coverage rates of essential health interventions; wide disparities in geographic and socio-economic status, and a largely unregulated private sector providing poor quality health services[5]; and lack of community awareness and seeking timely care for health conditions.[6–8] The majority of maternal, newborn and child deaths

## INTRODUCTION

Pakistan's high maternal, neonatal and child mortality continues to hamper its overall growth and development.[1] The global



are preventable and treatable given the programmes are designed and delivered through an efficient public and private health system that takes into account the geographic and social structures.[9]

The health system and services operate through both public and private sectors in Pakistan. The public sector features a three-tiered health delivery system offering a range of essential public health interventions. The basic health units (BHUs) and rural health centres (RHCs) together with the community outreach programme that is, the Lady Health Worker (LHW) programme, provide core primary healthcare services; tehsil headquarter hospitals (THQs) and district headquarter hospitals (DHQs) serve as first level and second level referral facilities with ambulatory and inpatient care.[10]

An LHW is associated with the public health facility usually at the primary level and secondary level. Each LHW has a catchment population comprising of approximately 200 households.[11] They target specific groups such as families with children under five and married couple eligible for family planning. Their key functions include increasing awareness about contraception, MNCH and nutrition. They also provide treatment for common illnesses like diarrhoea and acute respiratory infections (ARI), and distribute contraceptives for family planning. Furthermore, they identify danger signs for illnesses and refer people from the community to the health facility.

Around 70% of the population uses private providers, with expenses often paid out of pocket. Private providers are largely unregulated, though provincial governments have recently set up regulatory healthcare commissions to reduce high levels of unqualified providers and initiated licensing of private facilities.[1]

A number of strategies have been employed at community and facility levels to strengthen the public health service delivery system in Pakistan and elsewhere to improve the MNCH indicators.[12] Such approaches have attempted to improve the quality of care through reviewing and updating training material[13]; conducting training and providing essential commodities to healthcare providers (HCPs) and community health workers (CHWs).[14 15] For example, job aids-led intervention to improve quality of antenatal counselling in the Zou/Collines regions of Benin showed positive impact on danger sign recognition, birth preparedness, clean delivery and newborn care.[16] Training of HCPs improved contact and quality across the continuum of care in Ghana.[17] Furthermore, capacity building initiatives for healthcare workers have resulted in improved care quality of MNCH services in Jigawa State, northern Nigeria.[18] Provision of job aids and training of LHWs in Gilgit region of Pakistan resulted in a significant reduction of 38% in perinatal mortality (PNM) rate.[19] Furthermore, birth kits with 4% chlorhexidine for application to the umbilical cord were provided to traditional birth attendants to prevent omphalitis, thereby resulting in 38% reduction in neonatal mortality.[20] A diarrhoea pack with oral rehydration salts (ORS) and amoxicillin was distributed. This was feasible and acceptable with an impact on the incidence of diarrhoea and reduced hospitalisation rates.[21]

Strategies to improve MNCH have also focussed on improving demand and supply side factors through community mobilisation activities and setting up of self-help groups. Formation of community groups in Magu district, Tanzania, to identify and resolve issues during pregnancy and childbirth resulted in an increased maternal health knowledge and positive behaviour changes among healthcare workers and community.[22] A behaviour change intervention with women's self-help groups was evaluated in rural India.[23] They found significant improvement in using contraceptive methods, had institutional delivery, practiced skin-to-skin care, initiated timely breastfeeding, exclusively breastfed the child and provided age-appropriate immunisation. Furthermore, community mobilisation intervention led by LHWs in Hala, Pakistan, resulted in a 21% reduction in stillbirth rate and 15% neonatal mortality rate over a 2-year period.[24] Community management of pneumonia through LHWs showed that they can successfully diagnose and treat severe pneumonia at home. The LHWs were able to effectively reach children with pneumonia where referral was difficult.[15] Community case management of pneumonia through LHWs with oral amoxicillin at home was tested in two districts of Pakistan and proved to be safe and effective for treatment of severe pneumonia through LHWs.[25]

Furthermore, there is evidence on interventions for strengthening health information system.[26] For instance, a simple data improvement intervention led to completeness and accuracy of mother-to-child transmission of the HIV data in KwaZulu-Natal, South Africa.[27]

Finally, it is evident from previous studies that scaling up interventions using delivery platforms to reach poor and rural populations through community-based strategies and their successful implementation can prevent 58% of an estimated 367 900 deaths (15 900 maternal, 169 000 newborn and 183 000 child deaths) and 49% of an estimated 180 000 stillbirths in Pakistan alone.[15 19–21 24 26 28]

The UeN initiative aims to improve the health, wellbeing and survival of women and children in Pakistan through piloting and implementing effective MNCH interventions at scale in eight high-burden districts spanning three provinces and targeted to reach more than 11 million beneficiaries. The UeN initiative's main goal is to scale up the evidence-based and cost-effective MNCH interventions within existing platforms at the community and facility level. Study goal, objectives and activities are detailed in figure 1. It is anticipated the UeN intervention scale-up in eight districts will reach approximately 1.7 million under-5 children, 1.5 million women of reproductive age and 8.6 million other civilians. The intervention will target 509 public health facilities (19 maternal and child health (CH) centres, 150 government dispensaries, 255 BHUs, 59 RHCs, 18 THQs, 6 DHQs and 2 civil hospitals), 500 private providers, 6518 LHWs and 300 CHWs. The primary hypothesis of the study is to assess

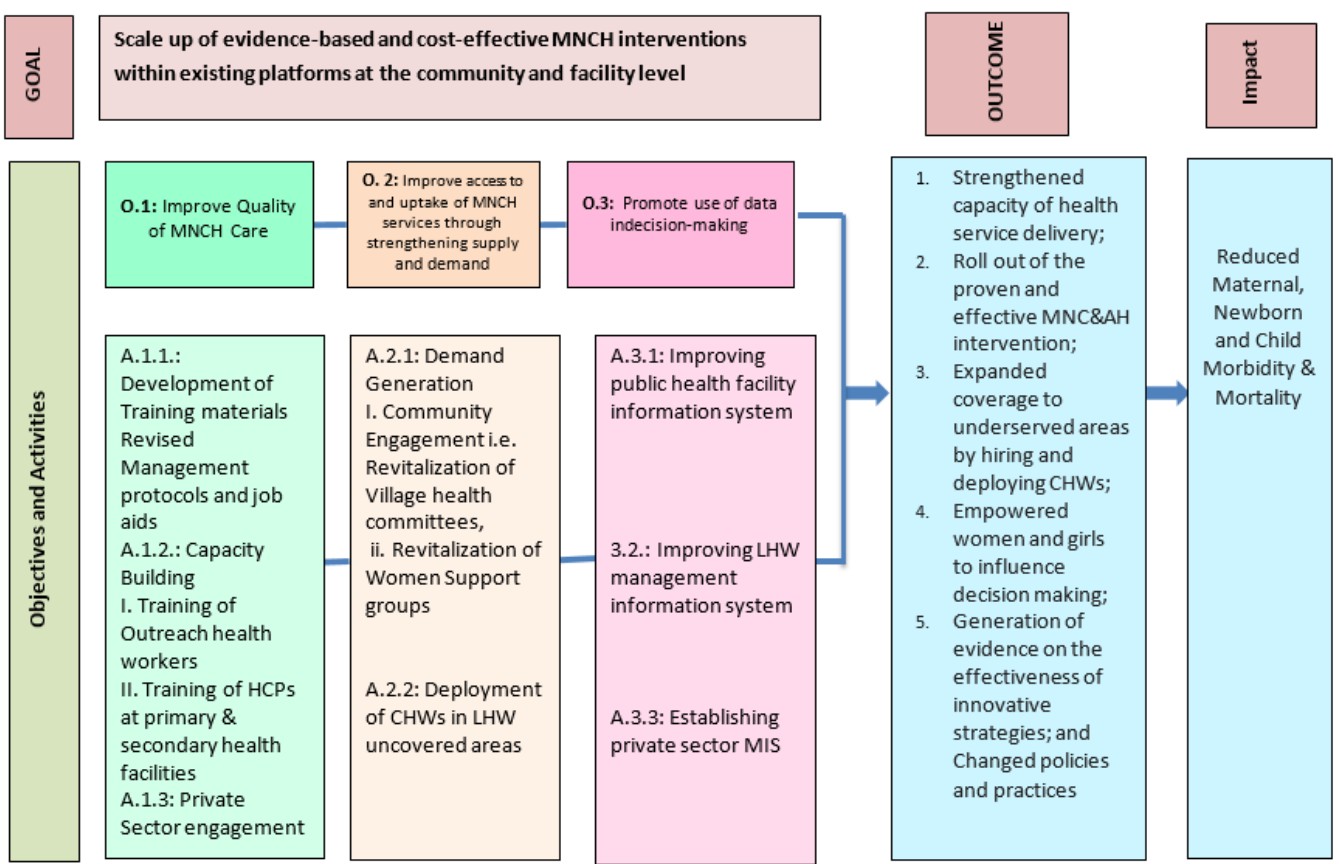

**Figure 1** Project design of Umeed-e-Nau. CHW, community health worker; HCP, healthcare providers; LHW, lady health worker; MIS, management information system; MNCH, maternal, newborn and child health; MNC&AH, Maternal, Newborn, Child and Adolescent Health.

whether roll out of proven and effective MNCH interventions at scale using existing public and private healthcare platforms at primary care level and secondary care level can reduce perinatal deaths by 20%, and diarrhoea and pneumonia case fatality rates by 30%.

## Objectives

1. Improve quality of MNCH care through development of training material and job aids for healthcare workforce; capacity building of healthcare workers; engaging private sector HCPs and ensuring supply of essential commodities at community and facility level.
2. Improve access to and uptake of MNCH services through targeting supply and demand side factors. The focus will be to create awareness on healthy practices at household level; revitalising and creating village health committees; and expanding community-based services to areas, which are currently not covered.
3. Promotion of the use of data for decision-making. Interventions will be used to improve data quality of public health facility and community level information systems. Furthermore, information system will be established for private HCPs.

This protocol paper will present the design and roll-out strategy for the UeN initiative. We will outline the setting, methods, implementation strategies, and monitoring and evaluation processes of the project roll-out spanning years 2017 to 2022 (project timeline shown in figure 2).

## METHODS AND ANALYSIS
### Study setting
Covering three provinces in the country, the UeN scale-up component will be implemented in eight districts (location of interventions by theme is shown in figure 3). Study districts were selected to be predominantly rural, and had some of the highest MNCH morbidity and mortality outcomes in the country. An important consideration was also ensuring that the selected districts did not have other major public health campaigns to avoid conflation or dilution of UeN impact. High-burden districts were strategically selected to demonstrate effective change models in difficult geographies so as to maximise project impact on population outcomes. Table 1[29] outlines relevant demographic, health and contextual indicators for each study district.

### Study design
In order to evaluate the impact of scaling up MNCH interventions, a quasi-experimental study design will be used. Each of the eight districts will be treated as an intervention site, and will be grouped together broadly into two

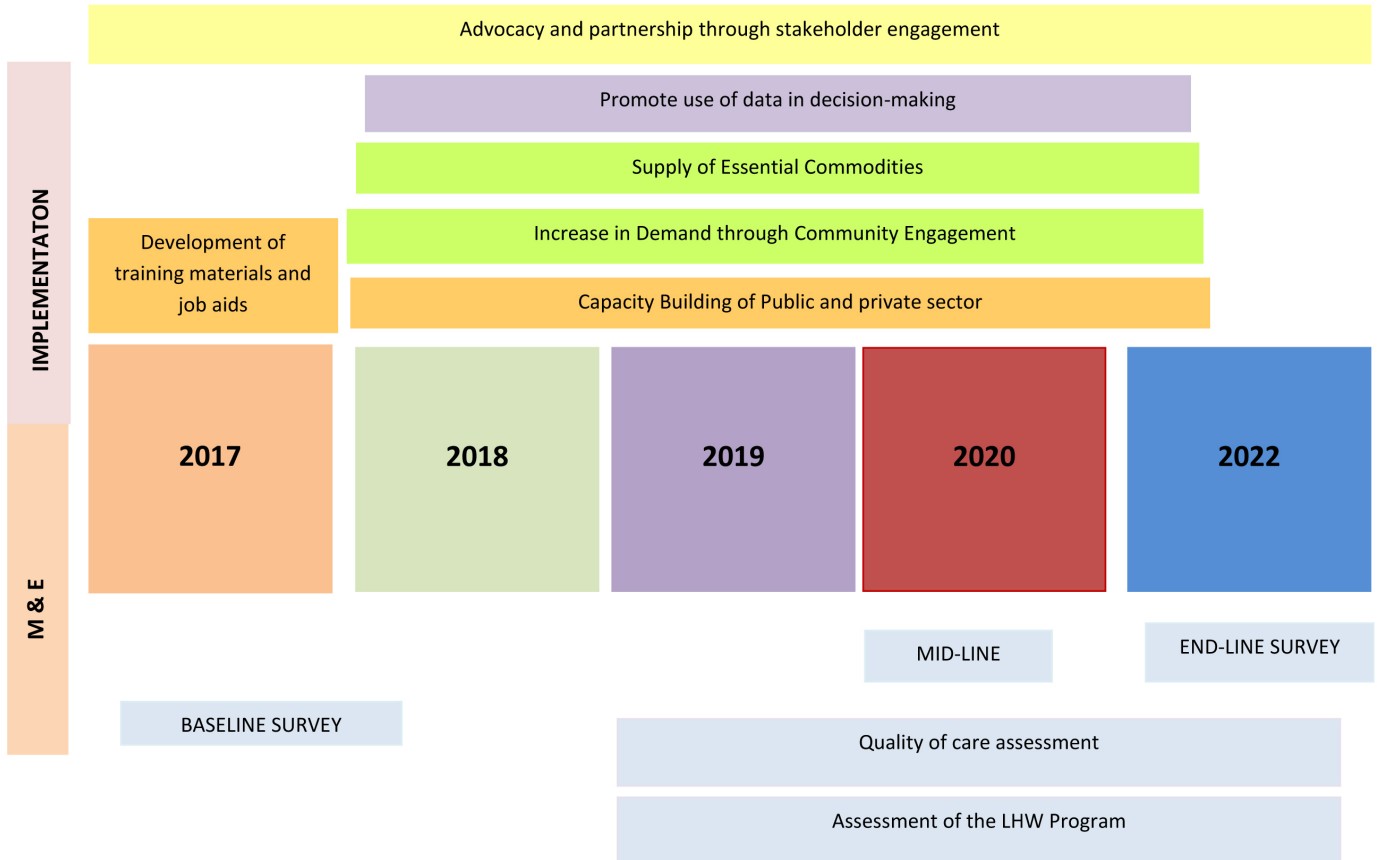

**Figure 2** Umeed-e-Nau project timeline. LHW, lady health worker; M&E, Monitoring and Evaluation.

areas to span MNCH interventions (figure 3). As detailed below, group 1 will receive maternal and newborn health (MNH) interventions, while group 2 will receive CH interventions. Each group will serve as the 'control' for the other. This design will help us evaluate independent and combined effect of MNCH interventions to address MNCH outcomes.

Site allocation to group 1 and 2 were done through extensive consultation with provincial government, bilateral donors and UN agencies, and UeN project team. Factors that were considered during intervention allocation included level of MNCH morbidity and mortality, assuring geographical diversity across the provinces, varying size of the districts and government MNCH priorities in those districts (so that those areas that were focused on improving CH, for example, received the CH interventions). This design might have ethical implications such as not providing all MNCH related evidence-based interventions to the entire population; however, population in control areas will continue to receive all the routine services which are part of the public sector standard of care and some additional MNCH intervention as per allocation to the type of intervention group. Furthermore, this ethical concern has been addressed by developing partnerships with Government, development agencies and private providers who have agreed to include UeN interventions in the existing health system,

if the evidence generated suggests their effectiveness and feasibility when delivered at scale.

### Sample size

The sample size for the household surveys will be calculated using the assumption that that the roll out of proven and effective interventions within existing health delivery platforms for MNCH will lead to significant reduction in PNM rate (20% compared with baseline) and at least 30% reduction in diarrhoea and pneumonia case fatality rates in the target districts for the proposed UeN scale-up initiative. PNM will be used as the primary outcome for all sample size projections. A stratified two-stage cluster sampling approach will be used to calculate sample size. Moreover, it is important to power the study to provide a detectable difference between the maternal and newborn health group, and the CH group. We will use a before and after design for sample size calculation. For MNH districts, assuming a 20% reduction in the PNM from the baseline PNM of 75 per 1000 live births (Pakistan Demographic Health Survey 2012 to 2013 for rural Pakistan)[30] with a power of 80% to detect the difference in the PNM between baseline, and the endline household survey, a type I error of 5%, design effect of 1.515 and an annual attrition of 10% per year, at least 19 582 households will be enrolled in the survey. For CH intervention districts, assuming a 30% reduction in the post neonatal death

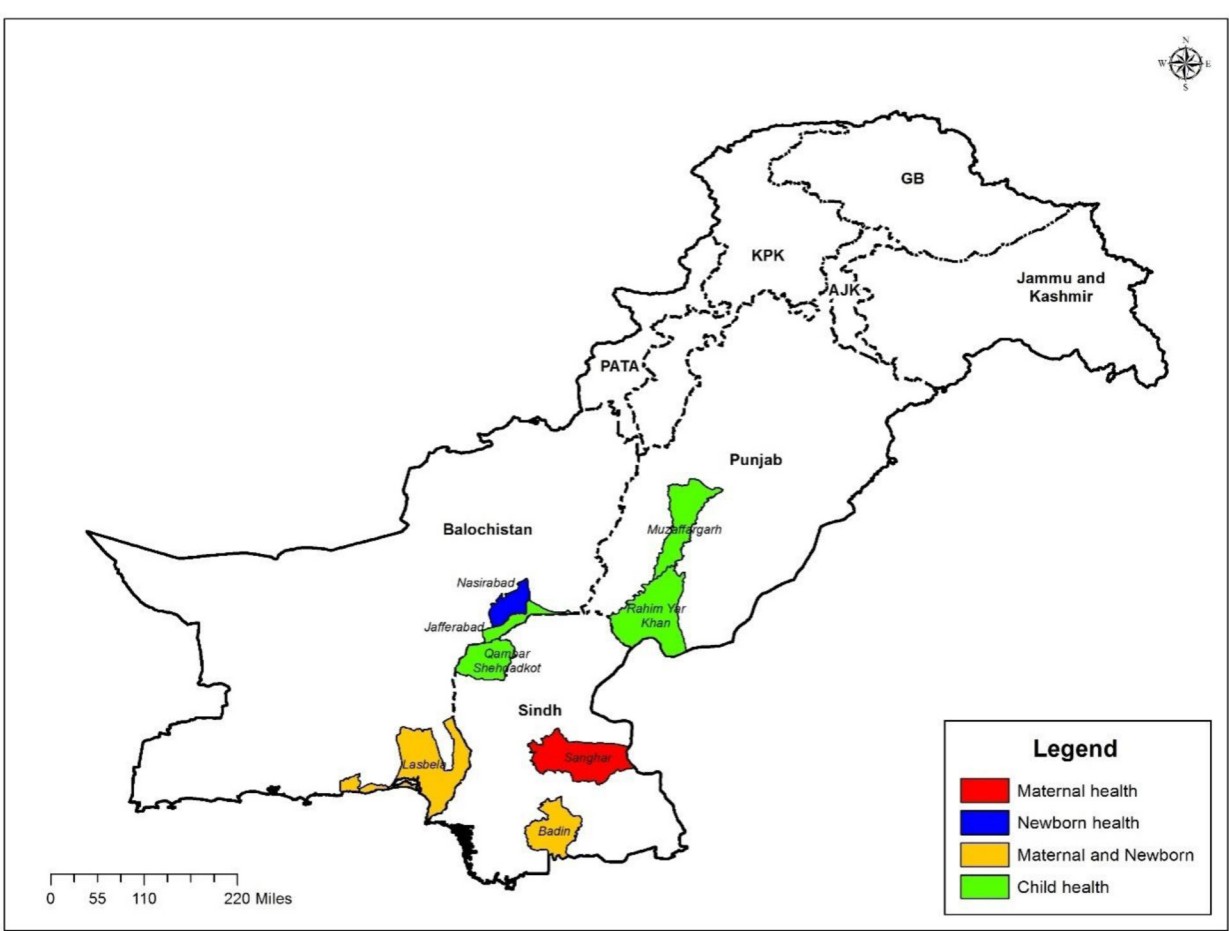

**Figure 3** Map showing the geographic location of the programme districts in three provinces of Pakistan. GB, Gilgit-Baltistan; KPK, Khyber Pakhtunkhwa; PATA, Provincially Administered Tribal Areas.

(PND) rate of 19/1000 live births at baseline (Pakistan Demographic Health Survey 2012 to 2013 for rural Pakistan) with a power of 80% to detect the difference in the PND between baseline and the endline household survey, a type I error of 5%, design effect of 1.388 and an annual attrition of 10% per year, at least 19 210 households have to be enrolled in the survey.

### Intervention packages and implementation strategies

An integrated intervention package, that has shown to be relevant, feasible and effective was designed thematically (figure 4) and will be delivered at scale in the target districts. Group 1: Maternal Health: These four districts (Badin, Sanghar, Lasbela and Nasirabad) will receive core MNH interventions as included in global recommendations for maternal health (Helping Mothers Survive)[31] and newborns (Helping Babies Survive)[32] consisting of three modules: Helping Babies Breathe, Essential Care for Every Baby and Essential Care for Small Babies. We stratified intervention delivery into three sub-groups focussed separately on maternal health (one district: Sanghar); newborn health (one district: Nasirabad); and combined MNH (two districts: Lasbela and Badin). This stratification was undertaken to independently evaluate the relative impact of theme-specific interventions on key

MNH and mortality outcome indicators; that is, akin to a three-arm trial design.

Group 2: Child Health: These four districts (Qambar Shahdadkot, Muzaffargarh, Rahim Yar Khan and Jafferabad) representing three provinces will receive CH interventions focussing on the implementation of the Global Action Plan for Pneumonia and Diarrhoea (GAPPD).[33]

Details on UeN intervention approaches packages are displayed in figure 1 and are discussed below by each of the five objectives.

### Objective 1: improve quality of MNCH care
#### Activity 1.1. development of training material and job AIDS

UeN will adopt and contextualise development of training material and job aids in accordance with global and national guidelines to build provider and health worker knowledge, skills and practices. The project team along with experts from the academia and government programmes will develop training packages (including trainer and learner materials, as well as supplemental job aids and guidance) for LHWs and for public and private HCPs.

For private HCPs, 14 modules for on-site learning using academic detailing to refresh the private providers' knowledge will be developed.

**Table 1** Distribution of demographic and health indicators for maternal, newborn and child health indicators in districts selected for programme implementation in UeN initiative in Pakistan

| | Maternal and newborn health | | Maternal health | Newborn health | Child health | | | |
| --- | --- | --- | --- | --- | --- | --- | --- | --- |
| | Lasbela | Badin | Sanghar | Nasirabad | Qambar Shahdadkot | Muzaffargarh | Jafferabad | Rahim Yar Khan |
| Population | 574 292 | 1 804 516 | 2 057 057 | 490 538 | 1 341 042 | 4 325 483 | 513 813 | 4 814 006 |
| Under-5 population | 74 657 | 202 106 | 230 390 | 13 837 | 150 197 | 462 756 | 14 191 | 625,8208 |
| HDI | 0.416 | 0.412 | 0.491 | 0.311 | 0.456 | 0.584 | 0.345 | 0.625 |
| HDI rank | 85 | 87 | 72 | 98 | 77 | 58 | 94 | 51 |
| Tehsils/talukas | 5 | 5 | 6 | 4 | 7 | 4 | 3 | 4 |
| Union councils | 22 | 12 | 63 | 31 | 52 | 93 | 38 | 139 |
| Vaccination coverage—Children 12 to 23 months (fully immunised) | 28.8 | 45.5 | 47.4 | 28.8 | 25.6 | 82.1 | 28.8 | 72.0 |
| Antenatal care during last pregnancy (%) | 55.9 | 83.1 | 56.8 | 55.9 | 64.5 | 37.4 | 55.9 | 30.8 |
| Facility births (%) | 34.6 | 64.8 | 40.0 | 34.6 | 48.7 | 50.7 | 34.6 | 62.0 |
| Delivery by skilled birth attendant | 38.2 | 66.6 | 42.1 | 38.2 | 49.7 | 52.7 | 38.2 | 65.6 |
| Postnatal consultation in health facility | 29 | 75.0 | 61.6 | 29 | 68.2 | 56.9 | 29 | 63.9 |
| Access to tap water | 29.6 | 20.7 | 20.7 | 28.6 | 5.8 | 6.5 | 28.6 | 21.4 |
| Access to toilet (flush) | 17.6 | 70 | 51.8 | 16.5 | 77.3 | 72 | 16.5 | 60.5 |
| Number of LHWs | 153 | 1010 | 883 | 309 | 680 | 1891 | 130 | 1723 |
| LHWs coverage % | 50 | 80 | 66 | 60 | 59 | 51 | 50 | 80 |
| Number of hospitals | 125 | 145 | 116 | 97 | 271 | 248 | 97 | 260 |
| Number of RHCs | 4 | 12 | 5 | 1 | 4 | 12 | 1 | 19 |
| Number of BHUs | 42 | 36 | 58 | 19 | 4 | 76 | 19 | 103 |

BHU, basic health units; HDI, human development index; LHW, lady health worker; RHC, rural health centres ; UeN, Umeed-e-Nau .

Group 1: Intervention Package                                                        Group 2: Intervention Package

**Sub-group 1 Implemented in one district: Maternal intervention package**

Capacity Strengthening (Public and Private Health Sector)

- Basic Emergency Obstetric care
- Comprehensive Emergency Obstetric care

Outreach Level

Capacity strengthening of LHWs in following:

- Importance of antenatal care
- Birth preparedness
- Importance of skilled birth attendance
- Identification of danger signs during pregnancy, labor, delivery and postnatal period Timely referral of maternal emergency
- Developing linkages between community based workers and providers

Sustained Supplies through government

- Provision of methyldopa
- Provision of MgSO4
- Provision of misoprostol
- Provision of clean delivery Kits

Deployment CHWs to provide maternal and newborn care in LHW uncovered areas

Community Mobilization and Education including:

- Formation/revitalization village Health committees to promote
- Maternal transport emergency mechanism
- Self-help groups for maternal care

**Sub-group 2: Implemented in one district: Newborn Health intervention package**

Capacity Strengthening (Public and Private Health Sector)

- Helping Babies Survive (HBS)
  - Helping Babies Breathe (HBB)
  - Essential care for every baby (ECEB)
  - Essential care for small babies (ECSB)
- Management of possible bacterial infections (PSBI)

Outreach Level

Capacity strengthening of LHWs in following:

- Importance of antenatal care
- Importance of skilled birth attendance
- Identifications of newborn danger signs
- Timely referral of newborn emergency
- Provision of clean delivery kits
- Provision of Chlorhexidine for cord care
- Thermoregulation of newborn
- Essential newborn care
- Promotion of early and exclusive breast feeding

Deployment CHWs to provide maternal and newborn care in LHW uncovered

Community Mobilization and Education

- Formation/revitalization village Health committees to promote newborn care
- Newborn transport emergency mechanism
- Self-help groups for newborn care

**Sub-group 3: Implemented in two districts: Maternal and Newborn Health intervention package**

Capacity Strengthening (Public and Private Health Sector)

- Basic Emergency Obstetric care
- Comprehensive Emergency Obstetric care
- Helping Babies Survive (HBS)
  - Helping Babies Breathe (HBB)
  - Essential care for every baby (ECEB)
  - Essential care for small babies (ECSB)
- Management of possible bacterial infections (PSBI)

Outreach Level: Capacity strengthening of LHW in following:

- Importance of antenatal care, Birth preparedness and skilled birth attendance
- Identification of danger signs during pregnancy, labor, delivery and of newborn
- Thermoregulation of newborn
- Essential newborn care
- Promotion of early and exclusive breast feeding
- Timely referral of maternal and newborn emergency

Sustained Supplies through government

- Provision of methyldopa
- Provision of MgSO4
- Provision of misoprostol
- Provision of clean delivery kits
- Provision of Chlorhexidine for cord care

Deployment CHWs to provide maternal and newborn care in LHW uncovered areas

Community Mobilization and Education

- Formation/revitalization village Health committees to promote maternal and newborn care
- Maternal and newborn transport emergency mechanism
- Self-help Groups for maternal and newborn care

**Group 2: Implemented in four districts: Child Health intervention package**

Capacity Strengthening (Public and Private Health Sector)

- General Danger Sign
- Case Management of Pneumonia and Cough or difficult breathing
- Case management of diarrhea
- Management of possible bacterial infections (PSBI)

Outreach Level

Capacity strengthening of LHWs in following:

- Identification of general danger signs
- Management of pneumonia and diarrhea at home
- Health education for WASH
- Immunization

Sustained Supplies through government/UNICEF/AKU

- Provision of ARI timer
- Provision of Amoxicillin DT for pneumonia
- Provision of Zinc +ORS for diarrhea

**Figure 4** Schema of intervention packages for the implementation of maternal and newborn interventions in Pakistan. AKU, Aga Khan University; ARI, acute respiratory infection; CHW, community health worker; LHW, lady health worker; ORS, oral rehydration salts; WASH, Water, Sanitation and Hygiene Education.

### Activity 1.2. capacity building

UeN will employ a mix of capacity-building activities to address both outreach workers' and public sector HCPs' performance and confidence in achieving quality standards. Training content will be based on the district's theme. Topics to be covered in the training are listed in figure 4.

► Outreach health workers training: LHWs

The LHWs' skills will be strengthened to promote healthy MNCH practices and behaviours at community-level and will be trained to adhere to care management protocols and use new commodities and technology which includes chlorhexidine, amoxicillin dispersible tablets (dt), zinc dt, low-osmolality ORS, respiratory counters and so on. The LHW training will be based on low-dose high-frequency model.[19 24]

► Facility-based: training for providers at primary level and secondary level public health facility

Training for public sector HCPs will be group-based where participants will gather in a public healthcare facility for a competency-based learning activity.

### Activity 1.3. private sector engagement

Private sector HCPs will be engaged through academic detailing, where individual health providers from private sector (or a facility-based team of providers) will participate in a structured learning activity at their facility. Topics on key areas of MNCH service delivery will be covered.

Initial interaction will be followed by facility-based coaching and mentoring.



### Activity 1.4. sustained key commodities and supplies by increasing efficiency of the supply chain for essential commodities at community and facility level

Stock Outs of MNCH commodities is a major bottleneck in delivering quality services within the health system.[34] LHWs, primary level and secondary level healthcare facilities in the target districts will be equipped with essential drugs and equipment to provide quality services through engaging officials of the department of health and other stakeholders. The commodities and supplies will be ensured according to district themes such as for MNH districts, clean delivery kit will be provided to LHWs serving in MNH districts.

For the four MNH districts, clean delivery kits will be provided by Aga Khan University (AKU). For the CH districts, supplies of amoxicillin dt (250 mg), zinc dt, ARI timer and pulse oximeter will be provided by UNICEF in three districts of Punjab and Sindh province; however, since the current UNICEF programme does not cover provision of commodities to Balochistan, AKU will supply the commodities to the CH district in Balochistan.

### Objective 2: improve access to and uptake of MNCH services and practices

#### Activity 2.1. demand generation

▶ Improve uptake of healthy practices at household level through community mobilisation

Through community mobilisation activities, the community will be informed by LHWs about danger signs during pregnancy, child birth, infancy and childhood. The importance of early referral by LHWs to health facilities for prevention of morbidity and mortality will be the focus of community mobilisation activities.

▶ Community engagement through revitalisation/ creation of women support groups/ village health committees

LHWs will be trained to revitalise and create where required self-help groups, village health committees and women support groups to discuss MNCH issues. These groups are expected to help women adopt healthy MNCH practices.

### Activity 2.2: expanding the coverage to underserved areas

In the current public health outreach system, LHWs provide services at the household level. However, only 50% to 60% of the rural districts are covered by LHWs.[35] To serve the geographic areas for which LHWs are not appointed, CHWs will be hired and trained on technical MNCH content, registration, community support group methodology and village health committee formation. Around 100 per district will be deployed. CHWs will be selected on the lower education qualification such as primary education or lower as compared with LHWs who are minimum middle pass. The CHW training will consist of lectures, interactive discussions and role-plays to develop the skills needed to perform their roles. The duration of the training will be 2 weeks, twice in the first year and then a refresher training every year. CHWs will

only be recruited and deployed in districts that have maternal, newborn or combined component due to current scope of community health worker programmes.

### Objective 3: promote use of data for decision-making

In Pakistan, there are two main sources of data for decision-making the District Health Information System (DHIS) and the LHW-Management Information System (MIS) for health facility and community level information, respectively. The private sector HCPs do not share routine service delivery data in the DHIS. To promote data use, UeN's objective is to target all three data sources. Initially, a gaps analysis will be conducted to identify the weaknesses in the system

### Activity 3.1. improving public health facility-related information system, that is, DHIS

The routine service delivery data of both primary level and secondary level health facilities is managed through DHIS system. The DHIS collects data on the 18 priority diseases, availability of drugs, service usage and human resource of health facilities. Data from each district is discussed in the quarterly district level meetings. However, there is poor compliance with the source registers which is the main source of information. As the data quality is poor and questionable, it cannot guide decision-making.[36] To improve understanding and importance of collecting data, HCPs of public health facilities will be trained on collection, recording and reporting of the data for DHIS.

For better compliance with source register and data quality, the project team will collect data from source registers on 40 key MNCH indicators every month. Data for these indicators will also be extracted from DHIS monthly reports from the online available dashboard. Values for the indicators from both source register and DHIS dashboard will be compared to assess data completeness and quality.[27] Monthly feedback report on data quality and completeness will be shared with the managers of health facilities and district managers. Furthermore, quarterly feedback report will be provided about the data quality and summary measures of key MNCH indicators in the district level management meetings.

### Activity 3.2 improving community-based health information system, that is, LHW-MIS

The LHW is affiliated with a primary health facility and has a catchment population of 1000 and maintains their records; for example, number of married couples, pregnant women, under-5 children and service provision on source registers provided by the LHW programme.[35] The information collected from the community level is compiled into a monthly report and shared with the lady health supervisor (LHS) and then relayed to the district coordinator for the LHW programme. However, the data quality is not monitored regularly and data use for decision-making is minimal. To improve the data quality, the LHS and LHWs will be trained on the importance of accurate data recording and reporting. Furthermore,

10 LHWs will be randomly selected from their reporting health facility every month; data from their source registers will be collected on 29 MNCH-related indicators on an Android application.

Values for the indicators from both source register and LHW-MIS dashboard will be compared to assess data completeness and quality. Monthly feedback reports will be shared with LHS and LHWs. Furthermore, quarterly feedback report about the data quality and summary of key MNCH indicators will be shared in the district level meetings.

### Activity 3.3. establishing private providers—MIS

A private provider MIS will be established. Routine data collection registers similar to the DHIS source registers with indicators according to the district themes will be developed and distributed to the private providers. Private providers will be encouraged to record data. The project team will collect data on monthly visits from the health facility. This will enable the capture of MNCH-related morbidities and mortalities which are otherwise lost due to non-reporting.

### Monitoring and impact evaluation

UeN scale-up will undertake rigorous monitoring and evaluation of core activities to ensure project progress and achievement of desired goals. Specific activities are detailed below:

► **Household surveys**

Household surveys will be conducted at three time points, baseline, at approximate mid-line when the interventions have been implemented at scale for at least 18 to 24 months, and at the end of the project (towards the end of year 5). The surveys will collect information to assess the impact of the interventions on primary outcome and secondary outcome indicators (listed in table 2).

► **Quality of care assessments**

Health facility assessment will be conducted bi-annually to provide information on facility level performance and service quality including availability of commodities and supplies, human resource and key equipment as well as observation of HCPs will be done using quality of care checklists to assess the effectiveness of training provided. In case of inadequate availability of resources or sub-optimal quality of care provided, arrangements for supplies will be done by liaising with stakeholders and refresher training will be provided to HCPs. All public health facilities (509, including 19 maternal and CH centres, 150 government dispensaries, 255 BHUs, 59 RHCs, 18 THQs, 6 DHQs and 2 civil hospitals) will be included in the study. For private providers, all health practitioners who provide services according to district theme will mapped. Practitioners with high patient flow will be included in the study for quality assessment and need-based support will be ensured.

► **Assessment of the community outreach programme (ie, LHW programme)**

The activities of LHWs will be monitored on a regular basis to strengthen monitoring system of LHW programme and improve LHW functionality. Thirty per cent health facilities will be selected on a quarterly basis, and 10 LHWs will be selected using purposive sampling from each health facility. Data will be recorded in a mobile application from the LHW's source registers for community support groups and 10 randomly selected households (*khandans*). The data for community support groups will be validated from the community and households will be visited to validate the household level data.

### Study outcome measures

The primary outcomes of interest are reduction in PNM rate (20% compared with baseline) and at least 30% reduction in diarrhoea and pneumonia case fatality rates. The secondary outcomes include coverage of essential intervention across the continuum of care and maternal, newborn and child morbidity rates (table 2).

### Analysis plan

#### Data management

A custom-made mobile application will be designed to collect data using Android handheld devices. The data collected in the field will be stored and accessible real time at AKU server. Android-based application will be designed such that it can monitor the location of the data points, time at which data was collected and time taken to fill each form. This will enable the data management unit consisting of IT specialists and statisticians to monitor data collection activities and quality. Furthermore, data cleaning will be done by identifying errors and missing entries before any analysis begins.

#### Statistical analysis

Basic frequencies will be calculated for demographic, social equity, primary outcome and secondary outcome of the study at baseline, and midline surveys and endline surveys. Furthermore, the baseline data will be analysed using multivariate analytical methods to assess the perinatal and child mortality estimates and its determinants.

The overall impact of the UeN project will be measured assuming parallel trends within the two MNH and CH groups, in the absence of any intervention, difference-in-difference estimates will be used to assess the impact of the intervention on outcomes. An interaction term between group (intervention/control) and survey wave (baseline/end-line) will be included in each model. All the analyses will be done using Stata.

### Ethics and dissemination

Ethics approval of the study was obtained from the Aga Khan UniversityEthics Review Committee on 28[th] February 2017. In addition, ethical clearance wassought from National Bioethics Committee (NBC), Pakistan.

Given the large scope of the UeN project, it is essential to garner support from and share the progress and findings of the study with the stakeholders. Therefore, stakeholder engagement activities will be conducted at



**Table 2** List of primary and secondary outcomes

| Outcomes | Definition |
|---|---|
| **Primary outcomes** | |
| Perinatal mortality rate | The number of stillbirths and deaths in the first week of life per 1000 total births |
| Diarrhoea case fatality | Per cent deaths among under-5 children due to diarrhoea |
| Pneumonia case fatality | Per cent deaths among under-5 children due to pneumonia |
| **Secondary outcomes** | |
| Maternal and newborn health | |
| Contraceptive use | Per cent of mothers aged 15 to 49 who are using modern contraceptive method |
| Antenatal care coverage | Per cent of mothers aged 15 to 49 who received antenatal care ≥4 times during pregnancy with their youngest living child aged <6 months |
| Tetanus toxoid coverage | Per cent of mothers aged 15 to 49 who received ≥2 doses of tetanus toxoid during pregnancy with their youngest living child |
| Skilled birth attendant at time of delivery | Per cent of mothers aged 15 to 49 whose youngest living child was delivered by skilled health personnel |
| Institutional delivery | Per cent of mothers aged 15 to 49 whose youngest living child was delivered in a health facility |
| Postnatal care coverage | Per cent of mothers aged 15 to 49 who received postnatal care (for self or for infant) within 3 days of birth |
| Early initiation of breastfeeding | Per cent of mothers aged 15 to 49 whose youngest living child was breastfed within 1 hour of birth |
| Exclusive breastfeeding | Per cent of mothers aged 15 to 49 whose youngest living child was exclusively breastfed |
| Vaccination coverage among under-5 children | Per cent of mothers aged 15 to 49 whose youngest living child received ≥3 doses of diphtheria, pertussis and tetanus/pentavalent vaccine when aged 12 to 23 months |
| | Per cent of mothers aged 15 to 49 whose youngest living child received ≥1 dose of measles vaccine when aged 12 to 23 months |
| | Per cent of mothers aged 15 to 49 whose youngest living child was fully immunised |
| Care-seeking for diarrhoea among under-5 children | Per cent of mothers aged 15 to 49 whose youngest living child with diarrhoea in the previous 2 weeks received oral rehydration salts and zinc |
| Care-seeking for pneumonia among under-5 children | Per cent of mothers aged 15 to 49 whose youngest living child with pneumonia-like symptoms in the previous 2 weeks was taken to an appropriate health provider and received antibiotics |
| Capacity building | Number of healthcare providers trained per 10 000 population |
| | Number of LHWs trained per 10 000 population |

LHW, lady health worker.

different levels to inform policy, community and district health authorities.

► **Participation in provincial child survival technical groups**

For reducing diarrhoea and pneumonia incidence, UNICEF has initiated the establishment of provincial child survival committees in which the UeN will participate. There are also existing provincial MNCH committees where AKU will have a seat by virtue of existing projects and the current UeN project. The UeN project will set up a system of monthly meetings with UNICEF and will share the project's progress and activities.

► **Formation of project advisory committee at national level**

Moreover, National Ministry of Health Services Regulation and Coordination will be requested to constitute a project advisory committee to help guide the UeN project through the existing federal ministry of health services and regulation to ensure coordination. The committee will meet once during the inception phase to discuss the project structure and remit, followed by bi-annual progress reviews to discuss challenges in implementation and address with appropriate actions.

► **Patient and public involvement**

The public was not involved in the design of the research tools but they will be a part of the study and their feedback will be sought regarding intervention acceptability. The key findings will be shared with their representatives as part of the dissemination plan at local level.

**Acknowledgements** We thank Mohammad Imran Khan, Rahat Qureshi, Shabina Arif, Ali Turab and their respective teams for their support in designing the intervention packages around maternal, newborn and child health.

**Contributors** ZB conceived the project and provided the technical and intellectual inputs as principal investigator in writing this manuscript and approved for submission. ZAM produced the first draft and subsequent drafts of the paper. NA, NK, SM, AH and SS were involved in review and provided inputs on various aspects of the manuscript. All authors reviewed and approved various drafts and the final paper.

**Funding** This work is supported by Bill and Melinda Gates Foundation grant number (OPP1148892).

**Map disclaimer** The depiction of boundaries on this map does not imply the expression of any opinion whatsoever on the part of BMJ (or any member of its group) concerning the legal status of any country, territory, jurisdiction or area or of its authorities. This map is provided without any warranty of any kind, either express or implied.

**Competing interests** None declared.

**Patient and public involvement** Patients and/or the public were not involved in the design, or conduct, or reporting, or dissemination plans of this research.

**Patient consent for publication** Not required.

**Provenance and peer review** Not commissioned; externally peer reviewed.

**ORCID iDs**
Zahid Ali Memon http://orcid.org/0000-0002-5321-8885
Sajid Soofi http://orcid.org/0000-0003-4192-8406

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
