## [Reviewer comments · BMJ Open]

ARTICLE DETAILS

TITLE (PROVISIONAL)	Effect and feasibility of district level scale up of maternal, newborn and child health interventions in Pakistan: a quasi-experimental study
AUTHORS	Memon, Zahid Ali; Muhammad, Shah; Soofi, Sajid; Khan, Nimra; Akseer, Nadia; Habib, Atif; Bhutta, Zulfiqar

VERSION 1 - REVIEW

REVIEWER	Mats Målqvist Uppsala university, Sweden
REVIEW RETURNED	27-Dec-2019

GENERAL COMMENTS	Dear authors, thank you for submitting your study protocol for review. This is a large population-based study with many components. The ambition is great and the interventions needed in the selected settings. Since this is a study that has already been started and is up and running, it is at this point not possible to alter the study design and the roll-out. I can therefore just give some comments for you to consider regarding future analyses and some discussion points. The first issue is on the research question. It is not clear from the manuscript what is the specific research question. A lot of evidence-based interventions are described, that are proven effective. Scaling-up of these is mentioned, but there is no clearly defined research question related to the scale-up. Is the purpose of the study to generate more evidence on the effectiveness of the selected interventions? Or is it to study mechanisms of scaling-up? If it is the first, the study design is quite blurred by the large number of health system strengthening measures included, which will hamper generalisations and replication. If it is the latter, methods for studying scale-up specifically are not mentioned. I would suggest to more clearly define what is intended to be studied, and how. To just have surveys analysing the primary outcomes and then make inference and say that changes were due to the UeN package is of little importance. Are the interventions equally effective when scaled-up as previously established effects in small studies, for example. But how would it be possible to single out such a question from a large package like UeN? Another issue with the study presented is on ethics. All interventions described are known to be effective. Yet the research team has chosen to apply a study design with control areas, withholding these
---

	interventions form half of the study population. One way to address this problem is to let all participants get some intervention, either MCN interventions or child health interventions. This is however not enough to get around the fact that the current study design is withholding interventions that are known to be effective from half the study population. Nothing is mentioned about a later implementation in the control areas, or a switch in interventions between the two areas. This ethical issue needs to be addressed in the study protocol.
--	---

REVIEWER	Onaedo Ilozumba Vrije Universiteit, the Netherlands
REVIEW RETURNED	03-Feb-2020

GENERAL COMMENTS	Overall, a very well elaborated protocol which describes a promising study with real societal relevance. There are a few points which could strengthen the manuscript:  -Authors should be sure to write all acronyms in full before using abbreviations eg LHW and UeN. Even if it was done in the abstract it should be repeated in the main text. -Given the significant involvement of LHWs in the project, a brief explanation of the cadre in the introduction would be helpful. - Page 4 -5 (lines 34-7): This paragraph could be restructured to improve comprehension. Currently it lists every intervention. However, reading the rest of the manuscript, it appears that these could be clustered in a way that links better to the project. For example, explaining LHWs and then providing the references of projects that they have been helpful with, etc. - Additionally, although there is a mention of successful interventions in Pakistan and elsewhere, but all references are from Pakistan. Have any of these interventions also been shown effective in similar contexts? - The objectives presented at the end of the introduction are quite broad and no-specific. It would be helpful to already bring the objectives that are expanded upon throughout the rest of the manuscript to the introduction. - Objective 4 and 5 as they are written are not actual objectives.eg objective 5: monitoring and evaluation -The description of baseline data cleaning and analysis is unclear. Exactly how will data cleaning and error monitoring be conducted? -The abstract misses a clearly articulated objective of the manuscript. Additionally, the manuscript is very rich and the abstract does not seem to present a comprehensive overview but rather focuses on smaller sections. For example 6 activities are highlighted as opposed to the 5 objectives presented. Also one hypothesis is presented in the abstract, however, from the manuscript it was not clear that this was the central study hypothesis.
---

VERSION 1 – AUTHOR RESPONSE

Reviewer 1 (comment 6): The first issue is on the research question. It is not clear from the manuscript what the specific research question is. A lot of evidence-based interventions are described, that are proven effective. Scaling-up of these is mentioned, but there is no clearly defined research question related to the scale-up. Is the purpose of the study to generate more evidence on the effectiveness of the selected interventions? Or is it to study mechanisms of scaling-up?

If it is the first, the study design is quite blurred by the large number of health system strengthening measures included, which will hamper generalisations and replication.

If it is the latter, methods for studying scale-up specifically are not mentioned. I would suggest to more clearly define what is intended to be studied, and how.

To just have surveys analysing the primary outcomes and then make inference and say that changes were due to the UeN packaged is of little importance.

Are the interventions equally effective when scaled-up as previous established effects in small studies, for example. But how would it be possible to single out such a question from a large package like UeN?

Author Response: Amended. "The primary hypothesis of the study is to assess whether roll out of proven and effective MNCH interventions at scale using existing public and private healthcare platforms at primary and secondary care level can reduce newborn deaths by 20% and child deaths due to Pneumonia and Diarrhea by 30%." This has been included before stating the study objectives in the introduction.

Reviewer 1 (comment 7): Another issue with the study presented is on ethics. All interventions described are known to be effective. Yet the research team has chosen to apply a study design with control areas, withholding these interventions from half of the study population. One way to address this problem is to let all participants get some intervention, either MCN interventions or child health interventions. This is however not enough to get around the fact that the current study design is withholding interventions that are known to be effective from half the study population. Nothing is mentioned about a later implementation in the control areas, or a switch in interventions between the two areas. This ethical issue needs to be addressed in the study protocol.

Author Response: Amended. An explanation has been added under the study design section, which reads as: "While this design might have ethical implications such as not providing all MNCH related evidence based interventions to the entire population however population in control areas will continue to receive all the routine services which are part of the public sector standard of care and some additional MNCH intervention as per allocation to the type of intervention group.

Furthermore, this ethical concern has been addressed by developing partnerships with Government, development agencies and private providers who have agreed to include UeN interventions in the existing health system, if the evidence generated suggests their effectiveness and feasibility when delivered at scale."

Reviewer 2(comment 8): Authors should be sure to write all acronyms in full before using abbreviations eg LHW and UeN. Even if it was done in the abstract it should be repeated in the main text.

Author Response: Amended. Now all acronyms have been defined in full at the first use.

Reviewer 2(comment 9): Given the significant involvement of LHWs in the project, a brief explanation of the cadre in the introduction would be helpful.

Author Response: Amended. Now a brief description of Pakistan's health system and LHWs has been included in the introduction. It reads as: "The health system and services operate through both public and private sectors in Pakistan. The public sector features a three-tiered health delivery system offering a range of essential public health interventions. The basic health units (BHUs) and rural health centers (RHCs) together with the community outreach program (i.e. the LHW program) provide core primary healthcare services; tehsil headquarter hospitals (THQs) and district headquarter hospitals (DHQs) serve as first and second level referral facilities with ambulatory and inpatient care.[14]

An LHW is associated with the public health facility usually a BHU or RHC. Each LHW has a catchment population comprising of approximately 200 households. They target specific groups such as families with children under 5 and married couple eligible for family planning. Their key functions include increasing awareness about contraception, MNCH and nutrition. They also provide treatment for common illnesses like Diarrhea and acute respiratory tract infections and distribute contraceptives for family planning. Furthermore, they identify danger signs for illnesses and refer people from the community to the health facility.

Around 75% of the population uses private providers, with expenses often paid out of pocket. Private providers are largely unregulated, though provincial governments have recently set-up regulatory health care commissions to reduce high levels of unqualified providers and initiated licensing of private facilities. "

Reviewer 2(comment 10): Page 4 -5 (lines 34-7): This paragraph could be restructured to improve comprehension. Currently it lists every intervention. However, reading the rest of the manuscript, it appears that these could be clustered in a way that links better to the project. For example, explaining LHWs and then providing the references of projects that they have been helpful with, etc.

Additionally, although there is a mention of successful interventions in Pakistan and elsewhere, but all references are from Pakistan. Have any of these interventions also be shown effective in similar contexts?

Author Response: Amended. The paragraph has been rewritten so that it aligns with the objectives of the study and includes examples of other countries also. It now reads as:

"A number of strategies have been employed at community and facility levels to strengthen the public health service delivery system in Pakistan and elsewhere to improve the MNCH indicators.¹⁰ Such approaches have attempted to improve the quality of care through reviewing and updating training material¹¹; conducting trainings and providing essential commodities to healthcare providers and community health workers^{12,13} For example, job aids led intervention to improve quality of antenatal counselling in the Zou/Collines regions of Benin showed positive impact on danger sign recognition, birth preparedness, clean delivery and newborn care .¹⁴ Training of HCPs improved contact and quality across the continuum of care in Ghana¹⁵. Furthermore, capacity building initiatives for health care workers have resulted in improved care quality of maternal, newborn and child health services in Jigawa State, northern Nigeria¹⁶. Provision of job aids and training of LHWs in Gilgit region of Pakistan had a significant reduction of 38% in perinatal mortality rate.¹⁷ Furthermore, birth kits with 4% chlorhexidine for application to the umbilical cord were provided to traditional birth attendants to prevent omphalitis and thereby effecting neonatal mortality resulting in 38% reduction in neonatal

mortality.¹⁸ A diarrhea pack with ORS and Amoxil distributed was feasible and acceptable with an impact on the incidence of diarrhea and reduced hospitalization rates.¹⁹

Strategies to improve MNCH have also focused on improving demand side factors through community mobilization activities and setting up self-help groups. Formation of community groups in Magu district, Tanzania to identify and resolve issues during pregnancy and childbirth resulted in an increased maternal health knowledge and positive behaviour changes among health care workers and community.²⁰ A behavior change intervention with women's self-help group was evaluated in rural India.²¹ They found significant improvement in using contraceptive methods, had institutional delivery, practiced skin-to-skin care, initiated timely breastfeeding, exclusively breastfed the child, and provided age-appropriate immunization. Furthermore, community mobilization intervention led by LHWs in Hala, Pakistan resulted in 21% reduction of in stillbirth rate and 15% neonatal mortality rate over two years period.²² Community management of pneumonia through LHWs showed LHWs can successfully diagnose and treat severe pneumonia at home. The LHWs were able to effectively reach children with pneumonia where referral was difficult.¹³ Community case management of pneumonia through LHWs with oral amoxicillin at home was tested in two districts of Pakistan and was proven to be safe and effective for treatment of severe pneumonia through LHWs.²³

Furthermore, there is evidence on interventions for strengthening health information system.²⁴ For instance a simple data improvement intervention led to completeness and accuracy of mother-to-child transmission of the human immunodeficiency virus data in KwaZulu-Natal, South Africa.”

Reviewer 2 (comment 11): The objectives presented at the end of the introduction are quite broad and no-specific. It would be helpful to already bring the objectives that are expanded upon throughout the rest of the manuscript to the introduction.

Author Response: Amended. The objectives have been brought at the end of the introduction. They now read as : “Objectives

1. Improve quality of MNCH care through development of training material and job aids for healthcare workforce; capacity building of healthcare workforce; engaging private sector healthcare providers and to ensure supply of essential commodities at community and facility level.
2. Improve access to and uptake of MNCH services through targeting supply and demand side factors. The focus will be to create awareness on healthy practices at household level; revitalizing and creating village health committees; expanding community based services to areas, which are currently not covered.
3. Promotion of the use of data for decision-making. Interventions will be used to improve data quality of public health facility and community based information systems. Furthermore, information system will be established for private healthcare providers.”

Reviewer 2 (comment 12): Objective 4 and 5 as they are written are not actual objectives.eg objective 5: monitoring and evaluation

Author Response: Amended. Objective 4 and 5 have been removed from objectives. Since objective 4 i.e. Encouraging Advocacy and Partnerships is a way to communicate the progress and findings of the study with the stakeholders, therefore it has been included in the dissemination plan now. And we agree objective 5 monitoring and evaluation is not really an objective. Therefore, we have removed it

as an objective but is still part of the manuscript because it shows how a large scale study such as UeN will be monitored and evaluated.

Reviewer 2 (comment 13): The description of baseline data cleaning and analysis is unclear. Exactly how will data cleaning and error monitoring be conducted?

Author Response: Amended. The section on analysis plan has been re-written with more details and clarity on data cleaning and data analysis. It now reads as: "A custom made mobile application will be designed to collect data using android handheld devices. The data collected in field will be stored and accessible real time at Aga Khan University server. Android based application will be designed such that it can monitor the location of the data points, time at which data was collected, and time taken to fill each form. This will enable the data management unit consisting of IT specialists and statisticians to monitor data collection activities and quality. Furthermore, data cleaning will be done by identifying errors and missing entries before any analysis begins.

Basic frequencies will be calculated for demographic, social equity, primary and secondary outcomes of the study at baseline, midline and endline surveys. Furthermore, the baseline data will be analysed using multivariate analytical methods to assess the perinatal and child mortality estimates and its determinants.

The overall impact of the UeN project will be measured assuming parallel trends within the two MNH and CH groups, in the absence of any intervention, difference-in-difference estimates will be used to assess the impact of the intervention on outcomes. An interaction term between group (intervention/control) and survey wave (baseline/end-line) will be included in each model. All the analyses will be done using STATA."

Reviewer 2 (comment 14): The abstract misses a clearly articulated objective of the manuscript. Additionally, the manuscript is very rich and the abstract does not seem to present a comprehensive overview but rather focuses on smaller sections. For example 6 activities are highlighted as opposed to the 5 objectives presented. Also one hypothesis is presented in the abstract, however, from the manuscript it was not clear that this was the central study hypothesis.

Author Response: Amended. The abstract now has all the objectives of the study. The hypothesis mentioned in the abstract has now been explicitly mentioned in the introduction. The Abstract now reads as: "ABSTRACT

Introduction

Pakistan has a high burden of maternal, newborn and child morbidity and mortality. Several factors including weak scale-up of evidence-based interventions within the existing health system; lack of community awareness regarding health conditions; and poverty contribute to poor outcomes. Deaths and morbidity are largely preventable if a combination of community and facility-based interventions are rolled out at scale.

Methods and analysis

The overall aim of the Umeed-e-Nau (UeN) project is to improve maternal, newborn and child health (MNCH) in 8 high burden districts of Pakistan by scaling up of evidence-based interventions. The

project will assess interventions focused on firstly improving the quality of MNCH care through a) development of job aids and training material; b) building Capacity of healthcare workforce; c) ensure provision of essential MNCH commodities; and d) engage with private sector health providers. Secondly targeting supply and demand side factors to improve access to uptake of MNCH services through a) creating awareness of healthy practices at household level; revitalizing and creating women groups and village health committees and expanding community based services to areas which are not covered. Thirdly by promoting the use of data for decision making through improving quality of public health facility and community health information system and establishing an information system for private sector providers. Furthermore, monitoring and evaluation of the program will be done through population level surveys, quality of care assessments at health facility level and assessment of the LHW program. The primary hypothesis of the UeN project is that roll out of evidence based interventions at scale will lead to a significant reduction in perinatal mortality rate (20% compared to baseline) and at least 30% reduction in diarrhea and pneumonia case fatality rates in the target districts where two intervention groups will serve as internal controls.

Household surveys will be conducted at three points (baseline, midline, and endline) to track progress and assess the impact of the UeN project. Descriptive and multivariate methods will be used for assessing the association between different factors and a difference in difference estimates will be used to assess the impact of the intervention on outcomes.

Ethics and Dissemination

The ethics approval of the study was obtained from the Aga Khan University Ethics Review Committee on February 28th, 2017. The progress and findings of the project will be shared with the provincial and national level stakeholders. Furthermore, results will be disseminated through open access peer reviewed journal articles.

This trail is registered, NCT04184544.”

VERSION 2 – REVIEW

REVIEWER	Mats Målqvist Uppsala University, Sweden
REVIEW RETURNED	23-Apr-2020

GENERAL COMMENTS	Thank you for responding well to previous comments. I wish you luck in the execution of this large-scale research project and will follow the progress with interest.
---